# Drug Potency Prediction of SARS-CoV-2 Main Protease Inhibitors Based on a Graph Generative Model

**DOI:** 10.3390/ijms24108779

**Published:** 2023-05-15

**Authors:** Sarah Fadlallah, Carme Julià, Santiago García-Vallvé, Gerard Pujadas, Francesc Serratosa

**Affiliations:** 1Research Group ASCLEPIUS: Smart Technology for Smart Healthcare, Departament d’Enginyeria Informàtica i Matemàtiques, Universitat Rovira i Virgili, 43007 Tarragona, Spain; sarah.fadlallah@urv.cat (S.F.); carme.julia@urv.cat (C.J.); 2Research Group in Cheminformatics and Nutrition, Departament de Bioquímica i Biotecnologia, Universitat Rovira i Virgili, 43007 Tarragona, Spain; santi.garcia-vallve@urv.cat (S.G.-V.); gerard.pujadas@gmail.com (G.P.)

**Keywords:** virtual screening, graph autoencoders, graph regression, graph convolutional networks, neural networks, molecular descriptors, molecular potency, SARS-CoV-2, drug, prediction

## Abstract

The prediction of a ligand potency to inhibit SARS-CoV-2 main protease (M-pro) would be a highly helpful addition to a virtual screening process. The most potent compounds might then be the focus of further efforts to experimentally validate their potency and improve them. A computational method to predict drug potency, which is based on three main steps, is defined: (1) defining the drug and protein in only one 3D structure; (2) applying graph autoencoder techniques with the aim of generating a latent vector; and (3) using a classical fitting model to the latent vector to predict the potency of the drug. Experiments in a database of 160 drug-M-pro pairs, from which the ${pIC}_{50}$ is known, show the ability of our method to predict their drug potency with high accuracy. Moreover, the time spent to compute the ${pIC}_{50}$ of the whole database is only some seconds, using a current personal computer. Thus, it can be concluded that a computational tool that predicts, with high reliability, the ${pIC}_{50}$ in a cheap and fast way is achieved. This tool, which can be used to prioritize which virtual screening hits, will be further examined in vitro.

## 1. Introduction

Many efforts were made at the start of the COVID-19 pandemic to identify a drug that would stop the replication of the SARS-CoV-2 virus [1]. The main protease (M-pro) and the RNA-dependent RNA polymerase have been investigated as two major targets, and a drug for each target, nirmatrelvir (PF-07321332) [2] and remdesivir [3], has been approved by the European Medicines Agency and the U.S. Food and Drug Administration to treat COVID-19 [4].

Virtual screening (VS) and other computer-aided drug design techniques have been widely used to suggest new compounds that inhibit M-pro [5,6,7] and other SARS-CoV-2 targets [8,9,10]. A crucial component of drug research is drug potency expressed in terms of the amount required to generate an effect of a specific strength. The hit compounds suggested by a VS typically do not have enough potency to be used as drugs but may be the starting point for a process of hit optimization [11,12,13]. A prediction of a compound’s potency would be a highly helpful addition to a VS process. The most potent compounds might then be the focus of further efforts to experimentally validate their potency and improve them. Free-energy simulations, such as free-energy perturbation, have been used to accurately predict protein–ligand free energies [14]. Nonetheless, these methods require a great amount of computing power. Specifically, their application to calculate Δ$G_{bind}$ in VS require the use of supercomputer or cloud-computing resources (e.g., [15,16]).

Molecular graphs are an example of a very natural way to describe a set of atoms and their interactions [17,18,19]. A graph, in general, is a data structure depicting a collection of entities (e.g., atoms), represented as nodes, and their pairwise relationships, represented as edges. There is a growing interest in having graph-based techniques applied to machine learning [20,21]. This can be attributed to their effectiveness in visualizing and characterizing instances of data with complex structures and rich attributes [22], capturing the inter-relationships between a system and its components.

In this paper, we propose a computational method to predict the quantitative activity of potential non-covalent inhibitors of the SARS-CoV-2 Mpro, which is quantified by the corresponding $pIC_{50}$ (i.e., the negative log of the half maximal inhibitory concentration value when converted to molar). It is called ReGenGraph: Regression on Generated Graphs. The method’s input is the crystallographic pose of a compound at the catalytic site of M-pro, but docked poses could also be used. The crystallographic or docked pose is then treated as a molecular graph. We apply graph regression techniques based on graph autoencoders (GAEs) [17,23] to predict the $pIC_{50}$. The drug potency prediction is achieved in two steps. First, the 3D structure of the M-pro/drug complex is converted into an interaction graph, which represents the M-pro/drug complex as a whole structure. Then, the potency value is deduced through an autoencoder and a graph autoencoder. Since the reconstruction of the interaction graph is needed for learning purposes, we can visualize the reconstructed M-pro/drug complex and verify its quality. Figure 1 summarizes the scheme of the proposed approach.

In the next subsections, we briefly explain the main ideas behind autoencoders, which are the basics of our method.

### 1.1. Autoencoders

Autoencoders are a particular class of neural networks that are employed in machine learning to capture the most basic representations of an entity. To achieve this, they are trained to reconstruct the input data after having generated an intermediate data called latent space [24]. Autoencoders can be used for dimensionality reduction, data denoising, or anomaly detection. The obtained intermediate representations can also be used as learning tokens for classification and prediction tasks or for the generation of synthetic data.

An autoencoder consists of two components: an encoder that converts the input space into a latent space, resulting in a latent vector, Z, and a decoder that converts the lower-dimensional representation back to the original input space. We define $W_{0}$ and $W_{1}$ as the trainable weights in the encoder and decoder, respectively. The latent space, $Z\in R^{N\times a}$, is defined by the number of entities, N, (e.g., atoms in a molecule) and the features extracted in the latent space, a. Encoders and decoders include non-linear activation functions. This non-linearity typically increases the expressive ability of the network and enables it to learn a range of tasks at various levels of complexity.

### 1.2. Graph Autoencoders

There has been a growth in using neural networks on data represented as graphs across various domains, despite the complexity of graphs that results from their intertwined characteristics. For the scope of this work, we focus on applications concerning drug potency prediction [25]. The currently used techniques can be divided into four categories: recurrent graph neural networks, convolutional graph neural networks, graph autoencoders, and spatial–temporal graph neural networks [22].

A graph with attributes, represented by a node attribute matrix, X, and an adjacency matrix, A, can be represented as **G(X,A)**, where $X\in R^{n\times f}$ is a matrix of size n×f, with n being the number of nodes and f being the number of attributes. The adjacency matrix, $A\in R^{n\times n}$, is of size n×n, where $A_{i,j}=1$ if there is an edge between the *i*th and the *j*th node, and 0 otherwise. The graph’s edges are unattributed and undirected, meaning that if there is an edge from node *i* to node *j*, there is also an edge from node *j* to node *i*, which is represented by the equality, $A_{i,j}=A_{j,i}$.

GAEs are based on the concept of a graph convolutional network (GCN), which, in turn, is built on the notion of generalizing convolution-like processes on normal grids, e.g., images to graph-structured data through neural network layers [17].

The key idea behind GCNs is to define the neighborhood of a node in the graph using the information from the neighboring nodes to update the node’s representation. This can be accomplished by defining a convolution operation on the graph, which is typically implemented as a weighted sum of the representations of the neighboring nodes. A learnable weight matrix is often used to determine the weights of this sum, which the network learns as it updates the node’s representation [26]. Node attributes can also be used to infer global properties about the graph’s structure and the links between its nodes.

GAEs are composed of two main components: an encoder and a decoder. The encoder embeds input graphs through a GCN, as defined in [17], returning a latent matrix, $Z\in R^{n\times b}$, with the graph unique properties. The number of features in the latent space is b. Equation (Equation 1) shows the encoder’s function:(1)$$Z=GCN(X,A)=\tilde{A}ReLU\left(\tilde{A}XW_{0}^{\prime }\right)W_{1}^{\prime }$$
where $\tilde{A}$ is a symmetrically normalized adjacency matrix computed from A, while $W_{0}^{\prime }$ and $W_{1}^{\prime }$ are the weight matrices for each layer, which are learned through a learning algorithm. Note that ReLU is the classical non-negative linear equation.

The decoder is defined as Equation (Equation 2):(2)$$A^{*}=\sigma \left({\mathrm{ZZ}}^{T}\right)$$
where $\sigma \left(\cdot \right)$ is the sigmoid function and T means the transposed matrix. The output, $A^{*}$, is a matrix of real numbers between 0 and 1 that represents the probability of an existing edge in the reconstructed adjacency matrix. Note that, in order to deduce the final reconstructed matrix, a round function is applied to $A^{*}$ to discern between non-edge and edge, i.e., zero and one values.

As the aim of the GAE is to reconstruct the adjacency matrix such that it is similar to the original one, the learning algorithm minimizes the mean square distance between these matrices defined by Equation (Equation 3):(3)$$L=\frac{1}{n^{2}}\sum_{i=1}^{n}\sum_{j=1}^{n}w_{pos}A_{i,j}logA_{i,j}^{*}+w_{neg}\left(1-A_{i,j}\right)log\left(1-A_{i,j}^{*}\right)$$
where $w_{pos}$ and $w_{neg}$ are introduced to deal with the value imbalance between pairs of nodes with an edge and pairs of nodes without an edge.

## 2. Results and Discussion

A total of 160 M-pro crystallized structures bound to a non-covalent inhibitor for which its $pIC_{50}$ is known were used. Details about this database can be found in Section 3.1.

### 2.1. Molecule Reconstruction

The aim of this section is to give an example of the reconstruction of the ligand. As commented in the previous section, it seemed logical to think that a latent vector, Z, is representative enough of whether the system is able to return a good approximation of the ligand it comes from.

The adjacency matrix, $A^{*}$, produced by the GAE decoder (Equation (Equation 2)), is utilized to generate the bonds of the ligand. Note that the elements in $A^{*}$ are real numbers between 0 and 1. As a consequence, a link between atoms *i*, *j* was to be imposed if $A_{i,j}^{*}>0.5$ and no link otherwise. Additionally, the atomic number is reconstructed by the decoder of the autoencoder.

Figure 2 illustrates the ligand *Mpro-x0830* from the selected database and the compound generated by our method. It is evident that three chemical bonds, the edges in the graph, were not reconstructed properly. For future work, the imposed bond could be set to have a maximum length. As mentioned earlier, graph reconstruction is not the primary objective but rather the ability of the latent space to capture different graph structures by reconstructing them. In this sense, it could be the case that both the original and the reconstructed compounds produce almost the same latent vector despite not being identical compounds. Therefore, the fitting module might deduce similar properties, given the compounds are not identical.

### 2.2. Drug Potency Prediction

As mentioned in the Introduction section, a dual method was defined, in which the chemical compound composed of a drug and a protein is reconstructed through an autoencoder and a GAE. Since this is a novel method, the aim is to heuristically validate the need of using an autoencoder and a GAE instead of applying a classical scheme that is composed of only one of them, namely, an autoencoder or GAE.

Figure 3 shows three scatter plots of computed and experimental $pIC_{50}$ values corresponding to the compounds in the database. In the first case, only an autoencoder and a fitting function were used. That is, the module for encoding–decoding in Figure 1 was composed of a single autoencoder. Note that, in this scenario, the bonds of the compounds are not reconstructed. In the second case, only a GAE and a fitting function were used, meaning that the encoding–decoding mechanism in Figure 1 was composed of a single GAE. In this scheme, the compound can be reconstructed. Finally, our method was applied by combining latent representations derived from both the autoencoder and the GAE to be used by the fitting function.

The first technique returns the highest mean square error (MSE), with a value of 0.82, followed by the GAE technique where the MSE = 0.77, and, finally, the proposed method, with an MSE = 0.67. These outcomes validate the architecture in a practical example, which is based on splitting the node attributes—the features of the atom—into two parts: one that is independent of the graph edges—the existence and type of bond—while the other remained dependent on them. Hence, by carefully deciding which attributes should be taken into account and which ones should be discarded, it was demonstrated that it is worthwhile to define a dual model that applies this split of attributes.

As practical analysis confirmed the potential of our proposal, another set of experiments was carried out. To compensate for the small size of the dataset, this experiment was conducted with the “leave-one-out” method. That is to say, all the graphs were used for training while reserving one graph for testing. This process was repeated until all the data were used for both training and testing the model. The $pIC_{50}$ was predicted given the resulting vectors from the autoencoder, the GAE presented in [17], followed by the concatenated vectors from ReGenGraph (the proposed method).

The first column of Table 1 shows the mean of MSE and the standard deviation, given the semantic vectors resulting from the autoencoder applied to regression. The second column shows the same measures of the regression module applied to vectors obtained through a GAE, where the classical method was followed by using both semantic and structural knowledge without splitting. The third column shows the regression module applied to the ReGenGraph. The results indicate that there was a reduction in error when applying our approach in comparison to a classical one. Moreover, the standard deviation drastically decreased, which means ReGenGraph is less dependent on the data.

### 2.3. Runtime Analysis

The runtime for each training cycle varied from a few minutes to up to 40 min. **Technical specifications:** the experiments were conducted on a 2.4 GHz dual-core Intel Core i7 processor using Matlab R2022a.

## 3. Materials and Methods

The database is detailed in Section 3.1, our specific architecture is detailed in Section 3.2, and, finally, the learning algorithm is explained in Section 3.3.

### 3.1. SARS-CoV-2 M-pro Database

As mentioned previously, the dataset used consisted of 160 M-pro crystallized structures bound to an inhibitor for which its $pIC_{50}$ is known. A total of 53 of them came from the well-know Protein Data Bank (PDB) database, and the other 107 structures came from FRAGALYSIS [27] database. Table A1 in the appendix shows a list of the M-pro crystallized structures used for training the model.

Given a pair of ligand–protein, only one attributed graph was generated. This graph represents the whole ligand and only the atoms and bonds of the protein that are close to some atom of the ligand, specifically at a distance lower than seven Å. Graph nodes are atoms of both the ligand and the protein. Graph edges represent bonds of both the ligand and the protein. Attributes on the nodes represent the three-dimensional positions of the atoms and their atomic number. Edges are unattributed, and there is an edge if there is any type of bond between atoms. The maximum number of atoms in the compound composed of “ligand + binding site atoms” is 146, and for this reason, all the generated graphs have 146 nodes.

As an example, Figure 4 shows the ligand at the x0689 FRAGALYSIS entry with (right) or without (left) the binding site environment. Note that only the parts of the Mpro with a distance smaller than 7 Å to the ligand are displayed, which is the part used for our purposes.

### 3.2. Architecture Configuration

The basis of the GAE approaches is the constraint that knowledge associated with nodes is related to knowledge attached to edges and vice versa [17]. That is, it is assumed that there is a relationship between the local structural pattern and the node attributes. In our case, the node attributes consist of the three-dimensional position of the atom and its atomic number. In the case of the first attribute, one can observe a clear relationship between having a bond, an edge in the graph, between two atoms, two nodes in the graph, and the proximity between these atoms. Contrarily, in the case of the second attribute, there is no relationship between the type of atom and being connected to a similar one. The contrary option would be, for instance, that oxygen tends to be connected to oxygen but not to other atoms.

The designed model was based on a GAE that handles graphs with nodes that have these two types of attributes: those that are impacted by structural patterns and those that are not related to edges. Specifically, our approach is based on two modules that work accordingly. The first one is an autoencoder [28] that captures semantic information, i.e., atomic number, without structural relations but rather by only utilizing certain node attributes. The other module is a GAE [17] that captures structural knowledge, i.e., atomic three-dimensional position, which is achieved by exploiting the remaining node attributes and edges. Both modules project their data into a latent domain, which is then used for any fitting mechanism, as shown in Figure 5. The GAE architecture defined in [29] and summarized in Section 1.2 was used. It is important to note that both the autoencoder and the GAE are used for extracting features in the encoder stage that can be used in a prediction or classification model. Nevertheless, whole models and encoder and decoder stages are also useful for reconstructing the graph.

The decision on which node attributes to use in the autoencoder and which to use in the GAE is made through a validation process. This can involve randomly selecting attributes for each architecture and determining the combination that results in the lowest loss for both. However, in specific problems, the user can make this decision based on their knowledge of the problem.

The latent space of the proposed architecture is created by combining the latent space of the autoencoder, represented as $Z_{\mathrm{sem}}$, and the latent space of the GAE, represented as $Z_{\mathrm{str}}$. Graphs are structures that must be invariant to the order of the nodes, meaning they have the property of being node-position invariant. A common way to achieve this property is by computing the sum, mean, minimum, or maximum of each feature for all nodes. The choice was settled on calculating the mean, as it makes the architecture independent of the number of nodes. Applying this mean is commonly known as the global average pooling. Then, the given $Z_{\mathrm{str}}$ vector $r_{\mathrm{str}}$ is generated by computing their mean. Note that the length of the vector, $r_{\mathrm{str}}$, is independent of the number of nodes, *n*. This is an important feature because it means that we can fit the system with graphs that have different numbers of nodes.

Finally, the fitting module utilizes the concatenated vector composed of $Z_{\mathrm{sem}}$ and $r_{\mathrm{str}}$. This vector is used to determine the global property of the graph, which is in the current approach the drug potency.

The autoencoder was modeled with a fully connected neural network, which only has one hidden layer with 20 neurons, and the length of $Z_{\mathrm{sem}}$ is 20. The input and output layers have 146 neurons. $W_{0}\in R^{146\times 20}$ and $W_{1}\in R^{20\times 146}$. Moreover, the hidden layer used a sigmoid activation function, while the output layer used a linear function. The back-propagation algorithm was used for learning.

The input X of the GAE is composed of a matrix of 146 (number of nodes) times 4 (3D position + atomic number). Additionally, the input A of the GAE is composed of a square matrix of 146 times 146. $W_{0}^{\prime }\in R^{146\times 20}$ and $W_{1}^{\prime }\in R^{20\times 20}$. $Z_{\mathrm{sem}}$ is a matrix of 146 times 20, and thus, $r_{\mathrm{str}}$ is a vector that has a length of 20.

Finally, the fitting function is modeled by a classical regression. Thus, it receives a vector of 40 elements, composed of 20 elements from $Z_{\mathrm{sem}}$ and 20 elements from $r_{\mathrm{str}}$. It outputs only one real number that represents the $pIC_{50}$.

### 3.3. The Learning Process

The learning process was achieved in two steps. Initially, by both weights, $W_{0}$, $W_{1}$, of the autoencoder, and $W_{0}^{\prime }$, $W_{1}^{\prime }$ in the GAE are learned given all graphs $G^{g}$, where g=1,...,k. Following that, the regression weights are learned, given the returned latent vectors $Z_{\mathrm{sem}}^{g}$ and $Z_{\mathrm{str}}^{g}$ of all graphs $G^{g}$ in the training set, where g=1,...,k. For the scope of this paper and its application, we focus on GAEs. More on the learning process of the autoencoder and weights, $W_{0}$ and $W_{1}$, can be found in the original work [24].

GAEs (Section 1.2) were modeled to reconstruct only one, usually huge, graph. Thus, the aim of the learning process, which minimizes Equation (Equation 3), is to reconstruct this unique graph. In that case, Z would have to be defined such that it resembles the inherent properties of this graph. We are in a different scenario. We wish that all latent spaces, $Z^{g}$, generated by all *k* graphs $G^{g}$ are able to reconstruct their corresponding graphs $G^{*g}$, given only one GAE, i.e., the same weights for all the graphs. In this way, the minimization criterion was redefined as the sum of Equation (Equation 3) to represent the loss function of all *k* graphs in the dataset as expressed in Equation (Equation 4):(4)$$L=\frac{1}{k}\sum_{g=1}^{k}L^{g}$$
where
(5)$$L^{g}=\frac{1}{n^{2}}\sum_{i=1}^{n}\sum_{j=1}^{n}w_{pos}A_{i,j}^{g}logA_{i,j}^{*g}+w_{neg}\left(1-A_{i,j}^{g}\right)log\left(1-A_{i,j}^{*g}\right)$$
describes the loss function per each graph $G^{g}$.

## 4. Conclusions

Finding a fast and inexpensive method for predicting the potency of antiviral drugs against SARS-CoV-2 has been a cornerstone of research in drug discovery in the last two years. Given the experimental data on 160 M-pro/drug non-covalent complexes, this aim can be achieved by modern computational methods based on machine learning. A drug potency predictor of non-covalent ligand inhibitors was presented, which was based on two steps. The first part is the conversion of the ligand–protein complex into the interaction graph. The second is a new architecture composed of an autoencoder, a graph autoencoder, and a regression module. Additionally, a third step can be introduced to reconstruct the ligand, allowing one to visualize and evaluate the reconstructed compound.

A key aspect of our approach is the separation of the semantic and the structural knowledge of the compounds. The first is processed through the autoencoder, while the second is processed through the graph autoencoder. This main feature is independent of the application, which means that the proposed method could have different applications in other fields. The only important aspect to be considered is discerning between attributes that are dependent on the structure and attributes that are not.

Practical experiments show the ability of ReGenGraph to predict drug potency. In addition to that, they also show that the mean square error of the drug potency prediction using a graph autoencoder is larger than using our method.

In future work, we plan to test our proposal by using different architectures for the autoencoder and also to apply other fitting functions in the regression model, such as neural networks. Despite the simplicity of the chosen functions, the results are promising.

## Figures and Tables

**Figure 1 ijms-24-08779-f001:**
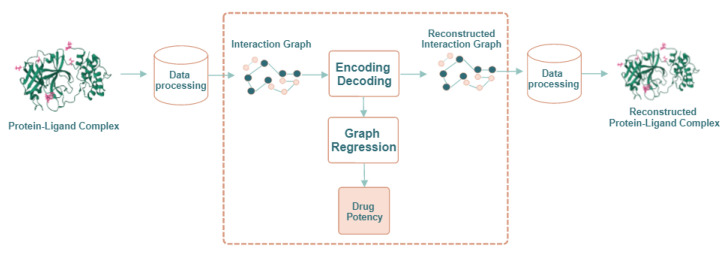
General scheme of our approach: ReGenGraph. The input is a protein–ligand 3D complex, while the output is their reconstructed structure and also the predicted drug potency.

**Figure 2 ijms-24-08779-f002:**
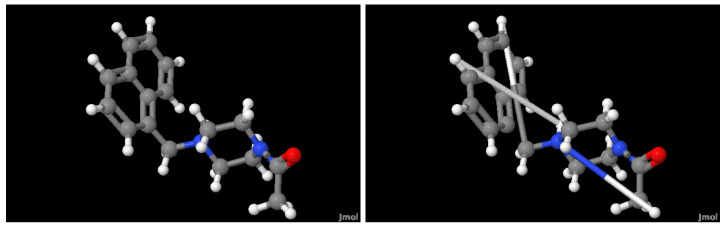
(**Left**) A ball-and-stick representation of ligand *Mpro-x0830*. (**Right**) The compound generated by our autoencoder and GAE. The ligand *Mpro-x0830* was randomly selected from the database.

**Figure 3 ijms-24-08779-f003:**
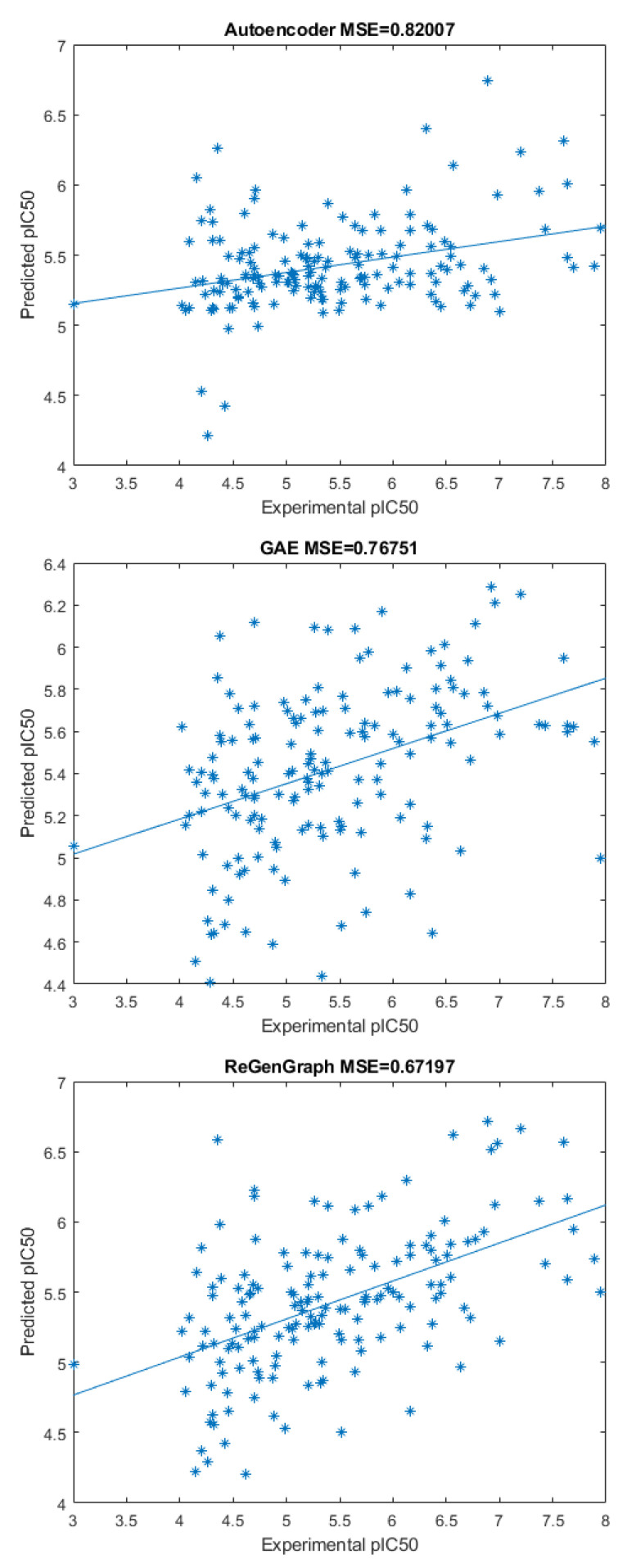
Three scatter plots showing the predicted and experimental $pIC_{50}$ of the compounds in the database. From top to bottom: using only an autoencoder, using only a GAE, and ReGenGraph (proposed model). The mean square errors appear on the top of the scatters.

**Figure 4 ijms-24-08779-f004:**
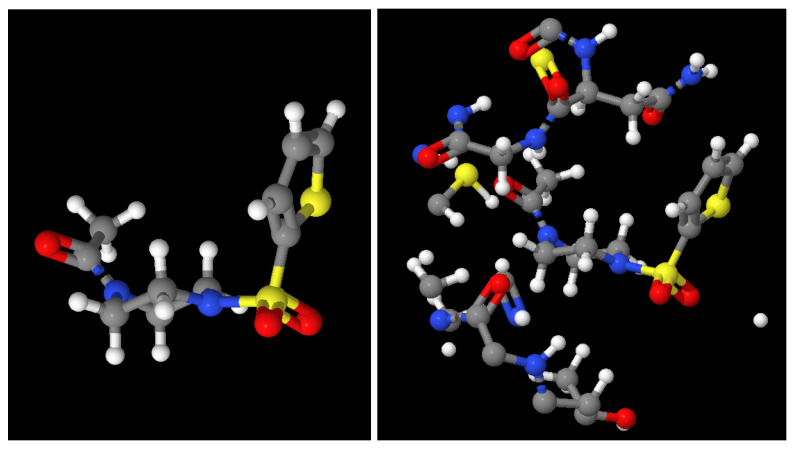
(**Left**): Ligand in complex x0689. (**Right**): Ligand and only the part of the Mpro close to the ligand (distance lower than 7 Å).

**Figure 5 ijms-24-08779-f005:**
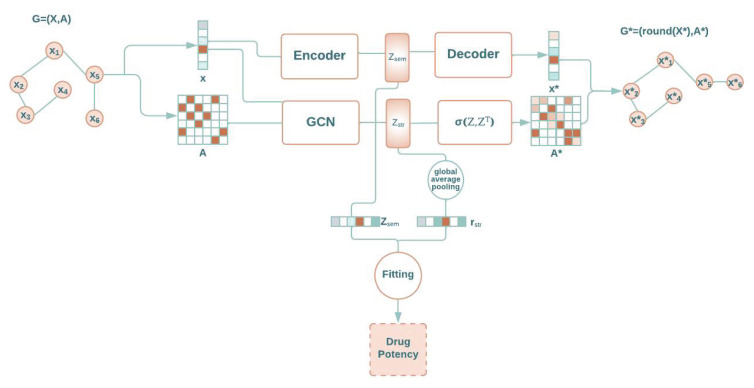
Schematic view of our architecture for graph regression based on an autoencoder, a graph autoencoder, and a fitting module.

**Table 1 ijms-24-08779-t001:** MSE and standard deviation obtained by an autoencoder, GAE, and ReGenGraph (our proposal).

	Autoencoder	GAE	ReGenGraph
Mean	0.83576	0.7456	0.6717
Std. Dev.	0.3188	0.9382	0.1796

## Data Availability

The scripts and data used for this experiment can be found on GitHub, https://github.com/ASCLEPIUS-URV/Mpro-complex-GAE, accessed on 5 May 2023.

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
