# Peer review of "Drug Potency Prediction of SARS-CoV-2 Main Protease Inhibitors Based on a Graph Generative Model"

_ijms, 2023, doi:10.3390/ijms24108779_

Round 1
Reviewer 1 Report
The authors propose a computational model based on regression on generated graphs for predicting the drug potency of non-covalent inhibitors of the SARS-CoV-2 Mpro. The manuscript is well-written, and the methodology is clearly explained. However, I have a few suggestions for the authors:
1. It would have been beneficial for the authors to validate their algorithm on multiple protein targets with a diverse range of shapes and symmetries.
2. In Figure 3, the protein-ligand complex names are not revealed. The authors could have included a list of the ligand-protein complexes used in this study in an appendix or supplementary text.
Author Response
Dear reviewer,
Thank you for your time and your interesting comments.
We answer each of your suggestions with a comment below.
- It would have been beneficial for the authors to validate their algorithm on multiple protein targets with a diverse range of shapes and symmetries.
Answer:
The main aim of our work is to build a predictive tool for the SARS-CoV-2 main protease because: (1) this is a target of great interest for COVID-19 infection treatment; and (2) there are several available 3D non-covalent complexes of this target with different drugs of known IC50 (enough to build a predictive model). Then, this predictive model has been tailored to predict the IC50 of other potential non-covalent inhibitors for this target. The same methodology could be used to build other target-tailored predictive models, but it would require an amount of experimental data (known 3D structures for non-covalent complexes and experimental IC50 values for the corresponding drugs) that it is not very easy to find for other targets.
Nevertheless, to consider your suggestion, we have slightly modified the last paragraph of the conclusions section. Thus, the following paragraph:
“In future work, we plan to test our proposal by using different architectures for the autoencoder and also to apply other fitting functions in the regression model, such as neural networks. Despite the simplicity of the chosen functions, the results are promising.”
has been modified to be:
“In future work, one the one hand, we plan to test our proposal by using different architectures for the autoencoder and also, to apply other fitting functions in the regression model, such as neural networks. Note that, despite the simplicity of the chosen functions, the results are promising. On the other hand, we intend to test our method with other protein targets, adapting the current architecture, if needed.”
- In Figure 3, the protein-ligand complex names are not revealed. The authors could have included a list of the ligand-protein complexes used in this study in an appendix or supplementary text.
Thank you for your suggestion. We have added a table in Appendix A with the names of the complexes.
Reviewer 2 Report
The manuscript entitled "Drug Potency Prediction of SARS-CoV-2 Main Protease Inhibitors Based on a Graph Generative Model" in which the authors defined a computational method to predict drug potency of SARS-CoV-2 Main Protease Inhibitors Based on a Graph Generative Model. They achieved a cheap and fast computational tool that predicts with high reliability the pIC50.
The work is understandable and the topic is important and appropriate for publication in International Journal of Molecular Sciences.

Author Response
Dear reviewer,
Thank you for your time. We have modified the paper according to the comments of the three reviewers.
Reviewer 3 Report
Title: Concrete, informative, adequate number of words.
Abstract: The reader is contextualized, the objective of the manuscript is clarified, the methodology is briefly reported, some results are reported, and some conclusions are presented. However, it is important that the abstract is written entirely in the third person.
Keywords: As a recommendation. Include the words: SARS-CoV-2, Drug, prediction.
Introduction: the authors carry out a good contextualization of the topic, the knowledge gap is clear, the objective and justification are well specified. It is recommended to change the background of figure 2 to a background of light colors.
Results and Discussion:
1. Lines 116, 120, 125, 140, 153, 158, 160, 161, 164, 173, 183, 204, 209, 217, 227, 231, 257 (We are in a different scenario), 263, 269, 277, 279 . Write in the third person
2. All figures must be right after being referenced.
3. "The learning process is achieved in two steps" write in past tense
Except for some minor corrections, according to my criteria, the manuscript has the academic-scientific quality to be accepted for publication.
Author Response
Dear reviewer,
Thank you for your time and your interesting comments.
We answer each of your suggestions with a comment below.
- Abstract:The reader is contextualized, the objective of the manuscript is clarified, the methodology is briefly reported, some results are reported, and some conclusions are presented. However, it is important that the abstract is written entirely in the third person.
Done, thanks.
- Keywords: As a recommendation. Include the words: SARS-CoV-2, Drug, prediction.
Done, thanks.
- Introduction:the authors carry out a good contextualization of the topic, the knowledge gap is clear, the objective and justification are well specified. It is recommended to change the background of figure 2 to a background of light colors.
We are using the Matlab viewer function and, unfortunately, we cannot modify the aspect nor the background colour of the figure. We are sorry about that.
- Results and Discussion:
- Lines 116, 120, 125, 140, 153, 158, 160, 161, 164, 173, 183, 204, 209, 217, 227, 231, 257 (We are in a different scenario), 263, 269, 277, 279 . Write in the third person
- All figures must be right after being referenced.
- "The learning process is achieved in two steps" write in past tense
We have rewritten the commented lines. Concerning the figures, we’ve written the text in latex, and it is the editorial compiler that imposes the position of these figures. Finally, we have written in past tense the sentence you commented.